# Depth-enhanced high-throughput microscopy by compact PSF engineering

Nadav Opatovski [1,6], Elias Nehme [2,3,6], Noam Zoref [2], Ilana Barzilai[2], Reut Orange Kedem[1], Boris Ferdman[1], Paul Keselman[4], Onit Alalouf[2] & Yoav Shechtman [1,2,5] ✉

High-throughput microscopy is vital for screening applications, where three-dimensional (3D) cellular models play a key role. However, due to defocus susceptibility, current 3D high-throughput microscopes require axial scanning, which lowers throughput and increases photobleaching and photodamage. Point spread function (PSF) engineering is an optical method that enables various 3D imaging capabilities, yet it has not been implemented in high-throughput microscopy due to the cumbersome optical extension it typically requires. Here we demonstrate compact PSF engineering in the objective lens, which allows us to enhance the imaging depth of field and, combined with deep learning, recover 3D information using single snapshots. Beyond the applications shown here, this work showcases the usefulness of high-throughput microscopy in obtaining training data for deep learning-based algorithms, applicable to a variety of microscopy modalities.

High-throughput (HTP) microscopy is instrumental in applications such as drug development and screening[1,2], study of cell processes[3–5], and treatment efficacy estimation[6] as these require testing different biological conditions in large quantities for statistical sufficiency. Furthermore, when incorporated with an incubator, these instruments allow the observation of multiple samples over prolonged observation windows (days to even weeks[1,7]).

Recently, anti-cancer drug screening and toxicology studies have widely adopted 3D multi-cellular tumor spheroids as a potential middle ground bridging the gap between 2D cell cultures and animal testing[8]. However, by large, ensemble analysis of these biological models with HTP microscopes is done over thin 2D slices, limited by imaging at a single focal plane.

The naïve solution for acquiring 3D information is to axially scan each sample at multiple focal planes by acquiring a z-stack[9]. However, this increases acquisition time and requires computationally demanding post-processing of 3D data. While sophisticated optical solutions such as high-throughput light sheet microscopy[10,11], or a focus-tunable lens[12] do provide axial information by axial scanning,

acquisition of 3D information remains time-consuming and is therefore often sacrificed for the sake of shorter experiment and analysis duration.

Point-spread function (PSF) engineering is a powerful microscopy technique that enables the extraction of typically unavailable information from biological specimens. In PSF engineering, the PSF, namely, the image that a point source generates on the camera, is modified using supplementary optical elements to encode information. This encoding, usually combined with computational image decoding, can enable 3D imaging[13–19], multispectral imaging[19,20,21], extended depth of field (EDOF) imaging[22–24], and more.

Most often, PSF engineering is implemented by extending the optical path of the microscope to gain access to the Fourier plane of the optical system, where wavefront shaping is performed[25]. This optical extension, in the form of a 4-f system, comprises of multiple elements, making it large and complex to align. In some microscope arrangements, size considerations are imperative, e.g., in HTP microscopes where the entire imaging apparatus moves laterally while scanning many samples per experiment or when the

[1]Russell Berrie Nanotechnology Institute, Technion - Israel Institute of Technology, Haifa, Israel. [2]Department of Biomedical Engineering, Technion - Israel Institute of Technology, Haifa, Israel. [3]Department of Electrical and Computer Engineering, Technion - Israel Institute of Technology, Haifa, Israel. [4]Sartorius Stedim North America Inc., Bohemia, NY, USA. [5]Department of Mechanical Engineering, University of Texas at Austin, Austin, TX, USA. [6]These authors contributed equally: Nadav Opatovski, Elias Nehme. ✉e-mail: yoavsh@technion.ac.il

microscope is placed inside an incubator. These considerations render PSF engineering by a 4-f extension practically impossible in such cases.

Here, we introduce a compact PSF-engineering modality implemented inside an Incucyte® S3 Live-Cell Analysis System (Sartorius BioAnalytical Instruments, Inc., Bohemia, NY). In our design, we position a phase-modulating element in the back focal plane (BFP) of the objective lens at its exit. This allows us to engineer the microscope's PSF and demonstrate applications such as single-shot z-projection and, additionally, combined with a deep learning analysis, snapshot 3D reconstruction of biological structures at high throughput and robustness to misfocus.

This work presents our implementation of compact PSF engineering in HTP microscopy. We showcase two cornerstone PSFs from the field of PSF engineering—an EDOF PSF[23,26] and the Tetrapod PSF[15]. The EDOF PSF maximizes the on-axis intensity over an extended axial range to provide an extended acquisition depth without the need for post-processing. The Tetrapod PSF is optimal for the 3D localization of individual emitters over a given axial range, thus providing 3D information via post-processing. We describe and analyze both PSFs experimentally, discuss the tradeoffs they exhibit, and through them, shed light on the opportunities in applying PSF engineering to HTP microscopy.

## Results

A microscope objective lens can be modified directly at its BFP, as has been demonstrated using amplitude masks[27]. In this plane, shift-independent changes to the PSF can be made by modulating the wavefront. In this work, we take advantage of the robustness of infinity-corrected objectives to the precise axial position of the phase mask to perform photon-efficient PSF engineering. In our implementation, the correct axial placement of the mask was validated by observing minimal field-dependent aberrations of the PSF (Supplementary Note 3).

We used two alternative modalities for mask placement at the exit of the objective lens. In the first, an annular mount was threaded into the objective turret, holding the phase mask in place just below the objective (Fig. 1A). In the second, a 3D-printed mask mount was attached to the bottom of the objective (Fig. 1B, Supplementary Movie 1).

### Direct snapshot extended depth of field imaging

Many HTP microscopy applications are limited by the native depth of field (DOF) of the system, while often a z-projection of the sample is desired rather than an image of a single z-plane[28]. Another DOF-related problem is the challenge of quickly focusing on each sample in an experiment, as they often have different planes of best focus. Both challenges can be addressed using an EDOF PSF.

Here, we implement EDOF imaging via PSF engineering. Our implementation does not require a deconvolution step, which alleviates any need for post-processing. The EDOF PSF provides a sharp image over an extended DOF by exhibiting greater robustness to misfocus than the standard PSF. Our implementation maximizes light throughput by using transparent phase elements, instead of incorporating amplitude modulation (e.g., NA reduction, see Supplementary Note 5).

Various EDOF methods that do not require an image deconvolution step have been previously investigated[22,23,26,29]. Here, we demonstrate two EDOF PSFs by a phase element placed at the exit of the objective. One element is composed by a photolithographically-fabricated phase mask, designed to maximize the depth of field of the PSF. The design process consisted of retrieving the BFP phase[30] that best approximates a constant Gaussian PSF, where photons are maintained in a tight single spot across an extended axial range[29]. The second EDOF element is a homemade multilayered glass element based on the principle of Abrahamsson et al.[26]. In short, EDOF is obtained by separating the BFP into several annular regions, which do not coherently interfere with each other. This way, the PSF comprises

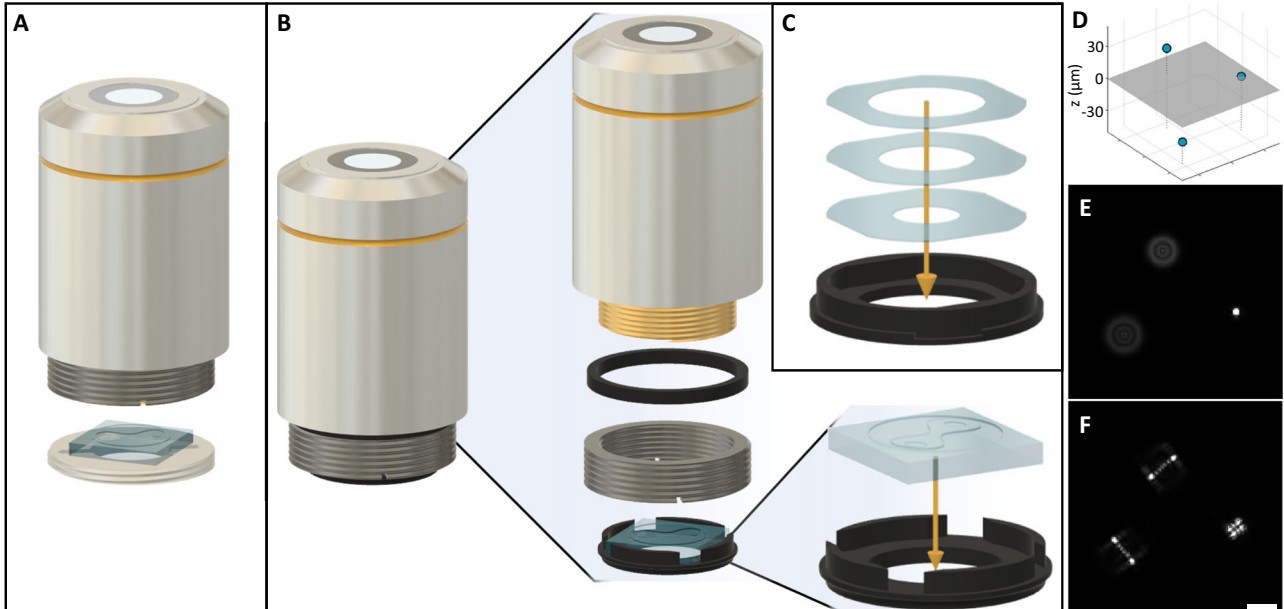

**Fig. 1 | Application of PSF engineering onto an HTP microscope objective.**
**A** Application external to the objective via a metal mount, onto which a phase mask is glued. **B** Application onto the objective using a 3D printed phase mask mount that attaches to the internal thread of the objective thread adapter. **C** Alternative EDOF design generated by stacked glasses rather than a phase mask. It is applied internally to the objective, similarly to the bottom element in panel **B**. **D** Depiction of three microspheres distributed in a 3D volume. Axial positions are +30, 0, and −35 μm, relative to the focal plane, marked in gray. **E** Simulated image of the microspheres from **D** by a standard HTP microscope. **F** Simulated image of the microspheres by an HTP microscope with a modified objective lens as shown in panel **B**. Lookup-table is the same as in panel **E**. Scalebar is 20 μm. The simulated PSF is a Tetrapod. A color-coded image of the phase pattern used is shown in Fig. 4A.

an incoherent sum of the PSFs produced by each of the regions. Additional description of the element is found in Supplementary Note 2 and in Chen et al.[24].

We tested the EDOF PSF with a synthetic sample of volumetrically scattered fluorescent microspheres fixed in gel and with fluorescently tagged nuclei of live spheroids. First, we evaluated the EDOF performance by imaging dense fluorescent microspheres suspended in gel. The bead density was ~22,000 mm$^{-3}$, resulting in ~2000 localizations per frame with the standard PSF, using ThunderStorm[31].

The standard PSF broadens quickly with defocus. As a result, the average localization precision is compromised. Moreover, PSF overlap becomes problematic in a dense environment, which further impairs localization quality. The EDOF PSF is far more robust to this issue, as the PSF remains compact over a long axial range before rapidly dispersing. We evaluated the EDOF contribution by finding the PSF width as a function of misfocus (Fig. 2B), as well as counting the number of localizations per width, presented in the histograms in Fig. 2C. By analyzing bead localizations within the entire volume, the EDOF mask exhibits an empirical improvement of the DOF by a factor of 1.9 (Supplementary Note 4). As a result, twice as many localizations were obtained with the EDOF PSF, with an average width (standard deviation of the Gaussian fit) of 1.39 µm. This is compared to 1.74 µm for the standard PSF (Fig. 2C). Often, sources that are laterally proximal but axially separated may not be imaged without suffering from an overlap of the defocused PSFs, unless the EDOF PSF is imposed. This is demonstrated in Fig. 2D.

Next, we performed EDOF imaging of fluorescently labeled nuclei in FaDu cell spheroids. Here, we used a photolithography-fabricated phase mask[30]. The results are presented in Fig. 3, showing the EDOF effect. Due to the characteristic spherical shape of the spheroid, each image exhibits a radial band of "best focus" around the spheroid center. This effect is more distinctive with the standard PSF, where the nuclei inside this band are very sharp, but the PSF divergence outside the band is swift.

The direct benefit of the EDOF PSF for biological sample imaging is that details can be observed over an increased depth, reducing the number of frames required to properly image a thick sample. Another advantage is relevant to high-throughput scanning experiments, where sample height at the imaging positions (e.g., the wells of a well plate) might vary. Today, this problem is addressed by a step of focus-finding prior to every image, which is time-consuming and may be destructive to the sample due to photobleaching. EDOF can relax the focus finding constraint as it tolerates some misfocus thanks to a decreased PSF sensitivity to axial change. The zoom-ins of Fig. 3 show the superiority of the EDOF PSF, as in a single image the object is sharply observable both at the center (spheroid bottom) and the side, compared to the single image with the standard PSF. We have also shown that further improvement can be attained through post-processing. We took advantage of the axially slow PSF change to apply Lucy–Richardson deconvolution[32,33] over the spheroid image, which significantly improves the results. More information can be found in Supplementary Note 1. An additional visualization is provided in Supplementary Movie 2.

## Snapshot three-dimensional imaging with CellSnap

In recent years, various methods have addressed the need for 3D imaging with enhanced temporal resolution, including lensless imaging[34], light-field imaging[35], PSF engineering[36], and more. Compared with PSF engineering, lensless and light-field imaging typically entail a degradation in lateral resolution, especially mask-only systems (such as a diffuser and a bare sensor), which have no magnifying optics and are thus limited to low effective numerical apertures (NA), although resolution, as well as reconstruction speed, can be significantly improved using appropriate post-processing[37,38]. Specifically, in the field of HTP microscopy, light-sheet imaging

with microfabricated culture chips[10] was recently proposed as an attractive candidate; however, this technique requires dedicated culture chips with embedded optical elements. Therefore, currently, there's an unmet need for a compact, scannable, easy-to-setup, and cost-effective HTP system that can rapidly capture and analyze 3D data.

Here we implement HTP 3D imaging using the Tetrapod PSF[15] (Fig. 4A, B), complemented with a deep neural network post-processing algorithm dubbed CellSnap. By combining the power of PSF engineering and deep learning, we demonstrate the possibility of 3D cell segmentation using a single snapshot, offering an order of magnitude faster acquisition compared to a traditional z-scan. Moreover, our approach is also computationally efficient, as our post-processing is highly parallelizable on a GPU and involves analyzing 2D images as opposed to 3D data cubes using existing methods[39,40].

To showcase our approach, we imaged live spheroids with fluorescently stained nuclei of the outer-shell cells, cultured in a 96-well plate (one per well, Supplementary Fig. 6). Training data consisted of pairs of z-stacks: one with the standard PSF and one using the Tetrapod PSF. We have taken advantage of the HTP capabilities of the microscope, enabling simple acquisition of large training datasets featuring hundreds of spheroids. Notably, such experiment-based training is superior to simulation-based training which exhibits limited consistency with experimental images when dealing with complex shapes and patterns. Z-stacks extending 400 µm were acquired—a necessary redundancy in range to counter variability in the axial position of the spheroids due to a global micro-tilt of the plate, as well as different in-well positioning of each spheroid (Supplementary Fig. 6). The variability in sample axial positions is inherent to HTP microscopes and serves as a further motivator for 3D imaging. Implications of this are discussed in detail in the Supplementary Information.

Next, we turn to describe our deep learning-based post-processing algorithm, dubbed CellSnap. To devise a neural network that handles a range of focal settings, we break the task into two parts. First, we train a focus finder that, given a snapshot taken with the Tetrapod PSF (shown in Fig. 4A, B), can estimate the focal plane at which it was measured (Fig. 4C). Afterward, we train a conditional 3D segmentation model (Fig. 4F), which, given a focus setting and a Tetrapod snapshot, can recover a canonical 3D segmentation of cell nuclei in the form of a binary occupancy grid, where occupied voxels belong to cells. For architecture details and learning hyper-parameters (see Supplementary Notes 8, 9).

The dataset for training CellSnap consisted of 592 spheroid "views" out of which 532 were used for training and 60 were used for validation (Supplementary Note 6). The model was then tested on another 20 spheroids from wells not seen during training/validation.

Training CellSnap consisted of two sub-tasks: training a focus finder and training a conditional 3D segmentation model, with each element in our dataset consisting of a pair of matched and aligned z-stacks acquired by scanning a spheroid twice: once with the Tetrapod PSF (Fig. 4C) and once with the standard PSF (Fig. 4E). First, the focus finder was trained on Tetrapod z-stacks alone (Fig. 4C). At each training step, a random Tetrapod snapshot was sampled, and the focus finder was trained to estimate its focal plane using the mean squared error (MSE) loss. The estimated focus was fairly accurate with an error standard deviation of ~4 µm (Fig. 4D), motivating us to train the conditional 3D segmentation model to output a voxel grid with an axial resolution of 4 µm.

For training the conditional 3D segmentation model, we used Cellpose[40] to segment the standard PSF stack and post-processed the result to produce a binary 3D segmentation label where occupied voxels belong to cells (Fig. 4E). The resulting stack-pairs for this phase of training were a z-stack of Tetrapod snapshots at different focal planes (serves as possible inputs), and the accompanying 3D cell segmentation at a canonical axial position (serves as desired output). At

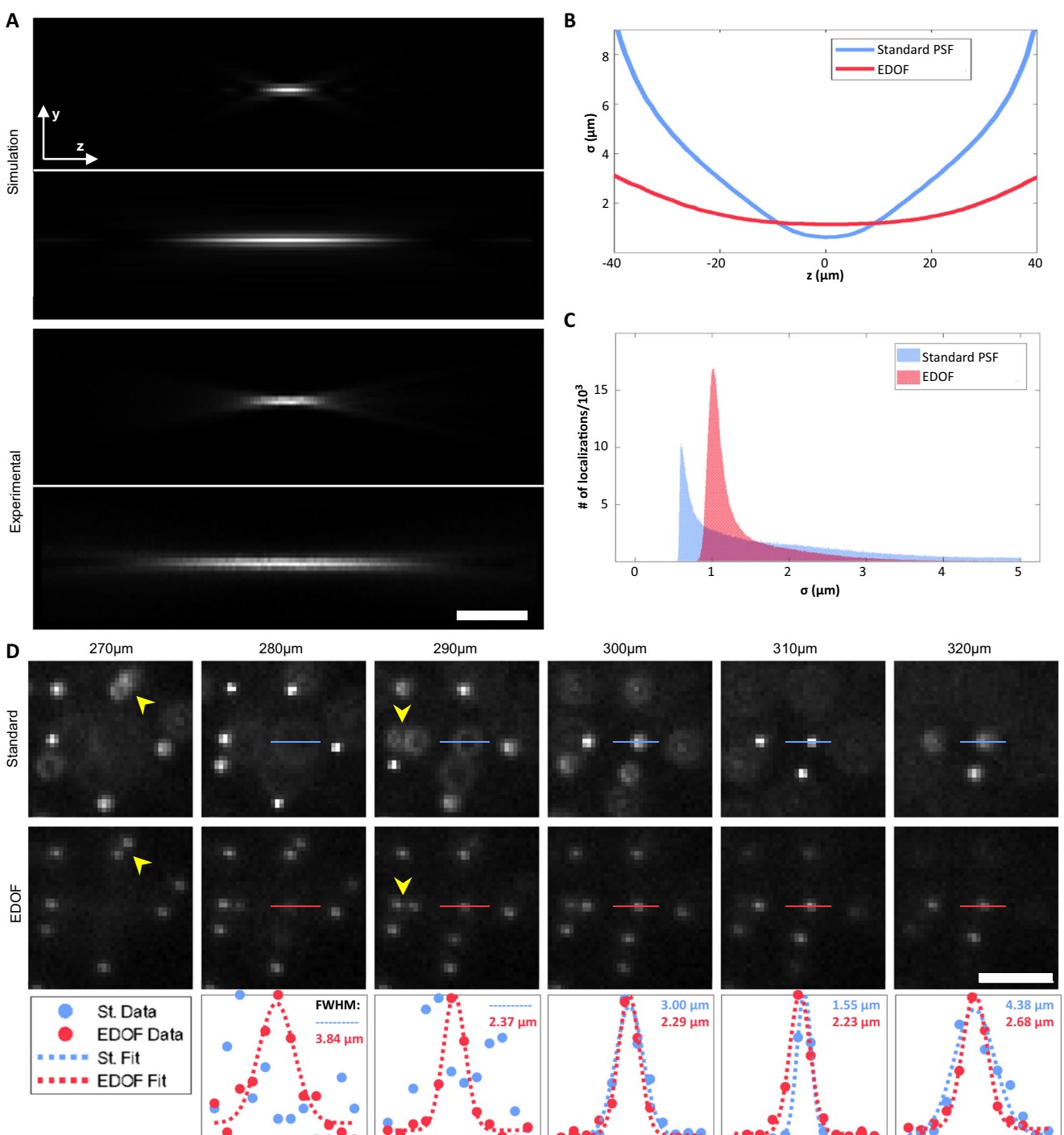

**Fig. 2 | Fluorescent bead imaging with EDOF. A** $y-z$ projection of the standard and EDOF PSFs, simulation and experimental. LUT is adjusted per projection. **B** Curves depicting the broadening of the PSFs as a function of defocus, generated from a statistical ensemble analysis. **C** Histograms of the number of localizations per PSF width, from the whole $z$-stack. **D** An ROI of the dense bead sample at various $z$-positions. Top row—standard PSF, middle row—EDOF. Yellow arrowheads mark beads that can only be colocalized with the EDOF PSF imposed. Imaging depth is specified above each column. The bottom row shows intensity profiles of the horizontal lines, with a Gaussian fit where applicable (the standard PSF profile of $z = 280, 290\,\mu m$ is not Gaussian). FWHM of the Gaussian profiles is specified near each plot, where FWHM $\approx 2.36\sigma$. In the legend, "St." stands for the standard PSF. All scale bars are $20\,\mu m$. Source data are provided as a Source Data file.

this stage, the focus finder parameters were kept fixed, and only the 3D segmentation model was trained using the Dice loss (Fig. 4F).

After both components of CellSnap are trained, at test time, we can feed a single Tetrapod snapshot at an arbitrary focal plane and obtain as output the 3D binary segmentation of cell nuclei. The results were also further post-processed using watershed splitting[41] to recover individual cell instances (Fig. 4G).

Figure 5 shows a representative example of CellSnap applicability. As a baseline, the results are compared to the application of Cellpose to a 3D z-scan of the standard PSF (Fig. 5A–C). Using a single snapshot with the Tetrapod PSF, CellSnap achieves 3D cell segmentation (Fig. 5D–F) at comparable quality to the baseline, with a slight performance deterioration at higher depths (edges in the image) where cells are densely packed (Fig. 5H, I). Importantly, compared to the

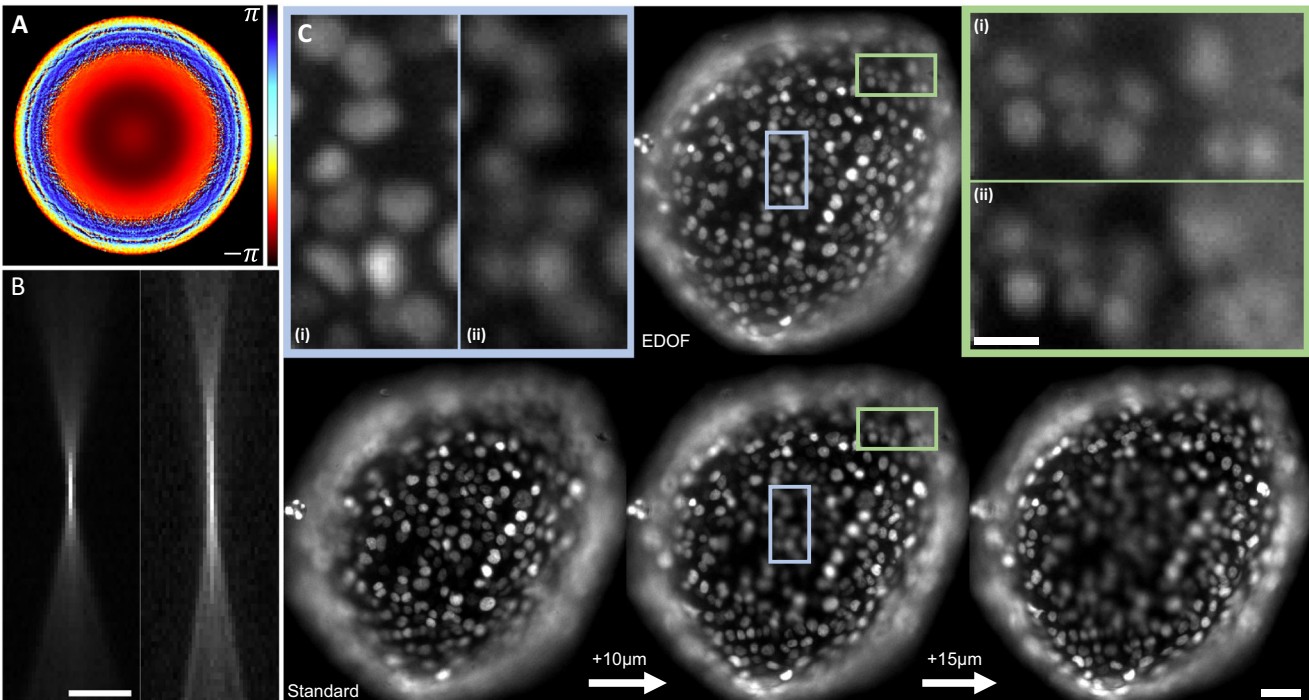

**Fig. 3 | EDOF imaging of a spheroid. A** Color-coded phase value of the lithography phase mask. Colorbar spans $2\pi$ radians. **B** $x$–$z$ projection of the standard PSF (left) and the photolithographically-fabricated EDOF PSF (right). Scale bar is 40 μm. **C** Top row (center)—EDOF image of the spheroid. Bottom row—three images of the spheroid with the standard PSF. Left—the lowest objective position in which the center is not blurred due to defocus. Right—the highest objective position that detects the outer nuclei that are captured by the EDOF image. Center—the best compromise between left and right. Scale bar is 50 μm. Zoom-ins (top row): (i) EDOF PSF, (ii) Standard PSF. Zoom-ins scale bar is 20 μm.

baseline, CellSnap was an order of magnitude faster both in acquisition (10×) and in post-processing (8.3×) (Fig. 5G and Supplementary Note 14).

When comparing the number of recovered cells as a rough estimate, CellSnap was able to roughly match the performance of Cellpose (Fig. 5J), with the results being highly stable across a broad range of focus settings (Fig. 5K), proving robustness to misfocus. In some cases, CellSnap was even able to recover cells missing from the segmentation label derived with Cellpose (Fig. 5B, E, H insets (i)). This is likely due to the robustness of deep networks to small amounts of label noise in training[42], as most training patches represent the correct mapping between 2D pixels and 3D cells (Supplementary Note 9). Nonetheless, such predictions were still mistakenly classified as false positives due to imperfect labels in the quantitative analysis described next (Supplementary Note 12).

We used two different metrics to quantify performance (Fig. 5L, M): the Dice coefficient and the "average precision"[39,40]. The Dice coefficient quantifies the per-voxel accuracy of our binary 3D segmentation, while the average precision highlights the performance on the individual cell level (Supplementary Note 11). The labels derived with Cellpose were mainly reliable at the lower part of the spheroids. Hence our quantitative analysis was restricted to the middle $150 \times 150$ pixels of the image (~120 μm axially), roughly capturing the bottom third of the spheroid. The reported quantitative metrics are the average of the entire test set over a large focus range of 140 μm. As a baseline for the quantitative analysis, we calculated the mean Dice coefficient and average precision for two random 3D segmentations in the test set. CellSnap provides strong segmentation results, highly correlated with the underlying 3D cells (see Supplementary Movies 3, 4). Supplementary Fig. 11 presents additional segmentation examples.

Finally, it is worth noting that even without PSF engineering, CellSnap alone can recover 3D information from standard snapshot data, although the performance is limited to a narrow focus range, significantly smaller than the range afforded by the Tetrapod PSF (Supplementary Note 13).

### Three-dimensional tracking with DeepSTORM3D

The main advantage of snapshot 3D imaging is the fast acquisition of volumetric information, which is particularly important for dynamic scenes. We demonstrate this advantage by using our high throughput microscopy for the application of *nanoparticle tracking analysis* (NTA)[43]. In NTA, particles diffusing in a suspension are tracked and their diffusion coefficient is extracted, which can provide information such as the diffusers' size distribution. Typical NTA studies make use of the standard PSF and perform tracking in 2D. Using PSF engineering can significantly enhance the information throughput of an NTA experiment by adding an extra tracking dimension over an extended z range. Our system can be seamlessly deployed for this task, combined with off-the-shelf software to handle data post-processing. To demonstrate this, we applied the Tetrapod PSF to the task of 3D single particle tracking of fluorescent microspheres diffusing in a water-glycerol mixture and analyzed the resulting time-lapse with DeepSTORM3D[14]. The results are presented in Fig. 6 (see also Supplementary Movie 5). Compact PSF engineering combined with DeepSTORM3D offers the ability to capture precise dynamics in 3D, a valuable capability that is rarely explored in NTA studies. Compared to a post-processing-only approach, combining DeepSTORM3D with the Tetrapod PSF increases the overall number of recovered tracks and the mean track length and substantially improves the localization precision in z (see Supplementary Figs. 12 and 13).

### Discussion

In this work, we introduced a simple, elegant way to implement PSF engineering in HTP microscopes. The approach opens the door to a variety of applications. We experimentally demonstrated two of the

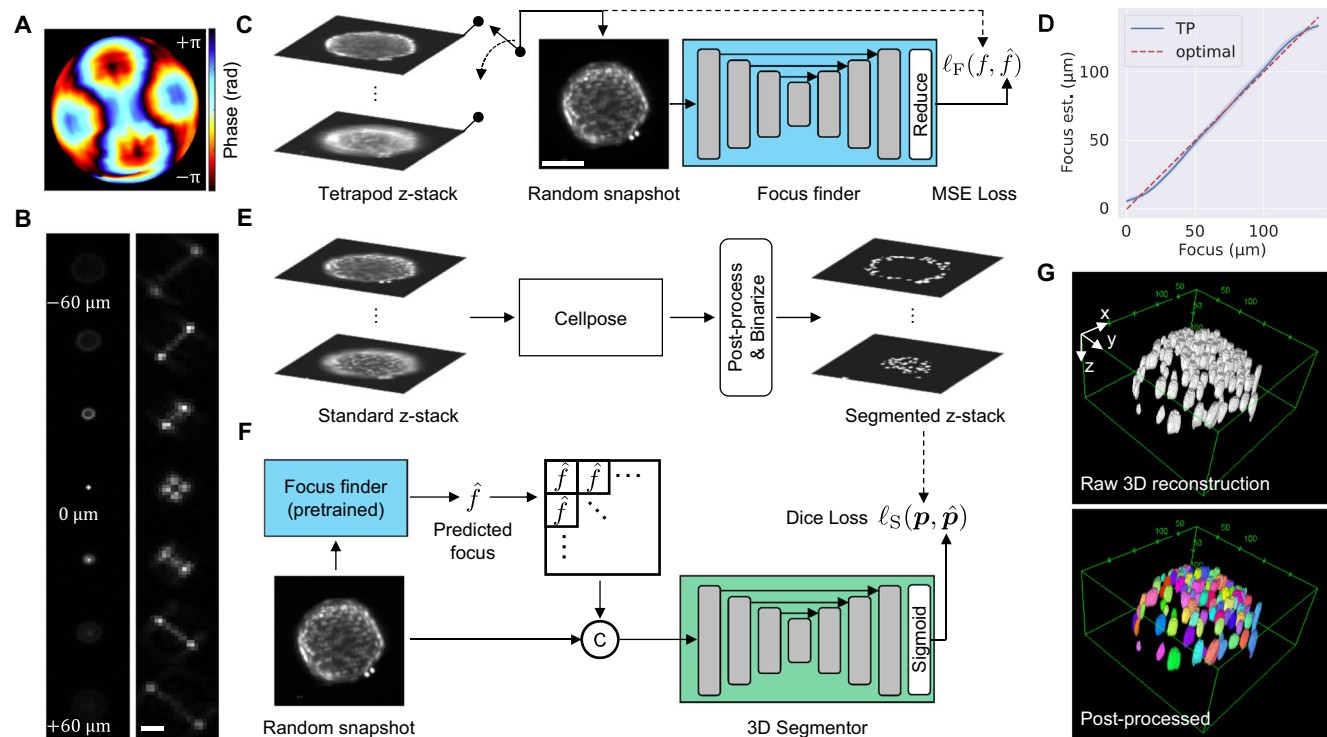

**Fig. 4 | Snapshot 3D segmentation with CellSnap. A** Tetrapod phase mask. **B** Standard (Left) and Tetrapod (right) PSFs as a function of emitter depth. Scale bar is 20 μm. **C** The Focus finder is trained to estimate the focus of a given snapshot with the Tetrapod PSF. Training is done using pre-processed z-stacks of spheroids imaged with the Tetrapod PSF. Scale bar is 100 μm. **D.** Estimated (blue) vs. Real (dashed red) focus for spheroids in the test set. Data is presented as mean value ± 95% confidence interval (i.e., 1.96*SEM). Estimation error standard deviation is 4 μm. Source data are provided as a Source Data file. **E** Accompanying standard PSF-z-stack is processed using Cellpose and the result is post-processed and binarized to produce a label for training the 3D Segmentor. **F** The 3D Segmentor is trained to predict a binary 3D segmentation given the Tetrapod snapshot concatenated with its predicted focus. **G** Raw binary 3D segmentation (top) and separate cell instances after post-processing with watershed splitting (bottom).

most common PSF engineering applications in an HTP microscope on synthetic and live samples. First, we presented EDOF imaging, i.e., a photon-efficient optical z-projection. The demonstrated implementations of EDOF could be suitable both for imaging and for performing localization microscopy using the Gaussian approximation of the PSF. The post-processing-free nature of the EDOF application is important because complicated analysis is often the bottleneck of elaborate microscopy techniques. Second, we demonstrated snapshot 3D cell segmentation using the Tetrapod PSF with a custom deep-learning architecture dubbed CellSnap. Exploiting this synergy between engineered optics and powerful computation enabled us to achieve comparable results to existing approaches while being an order of magnitude faster both in acquisition and in data post-processing. Notably, our proposed hardware modification is cost-effective, easy-to-setup, and highly customizable, paving the way to rapid and reliable 3D HTP microscopy. As another application of snapshot 3D imaging, we showcased the proposed hardware prototype with the Tetrapod PSF for NTA. Compact PSF engineering combined with DeepSTORM3D captured precise dynamics in 3D, improving z precision and increasing the depth range.

Incorporating PSF engineering in an optical system to increase the information throughput is subject to trade-offs. Most notable is the engagement with the optics, which in our work is done with minimal effort as an easily applied/removed add-on. Other drawbacks may include the requirement for post-processing, and resolution compromise, as we have presented in this work. Our proposed hardware-software combination triggers many possible questions regarding its capabilities and limitations. For example, on the optics side, how optimal is the Tetrapod PSF for imaging multiple cells rather than single point sources? It is likely that a content-aware PSF design[44], optimizing the "cell spread function", could provide a substantial boost in performance, as previously shown in dense multi-emitter imaging[18]. Similarly, on the post-processing side, how sensitive is our performance to the choice of the CNN architecture, output representation, loss function, and overall training schedule of CellSnap? Currently, it is unclear how each of these affects our bottom-line performance. However, the contributions presented in this work are orthogonal to efforts optimizing instance separation of crowded cells[39,40,45,46], and promoting model adaptivity to conditioning information[47,48] such as the focus setting. Hence, we envision that CellSnap performance can be boosted by adapting these advancements to our existing solution.

Finally, in this work we focused on the task of 3D cell segmentation as segmenting nuclei in microscopy images is typically the first step of any quantitative analysis performing phenotypic measurements on the single cell level[9]. However, our system can be tweaked on the post-processing end to address other interesting tasks encountered in HTP microscopy. For example, some applications might require the intensity profile of individual cells (e.g., when monitoring the activity of specific neurons). Other applications could be satisfied with a sample summarizing statistic, e.g., the number of dead cells after an external intervention. Clearly, our approach can be implemented with a variety of phase masks to fulfill any applicational requirement, including, e.g., spectral[19,21] or polarization[49] modulation. This work suggests that HTP microscopes can be made computational as facilitated by compact PSF engineering and deep learning, augmenting them with enhanced capabilities in a similar fashion to traditional microscopes, and possibly even more so considering the ease of obtaining large quantities of training data using HTP acquisition.

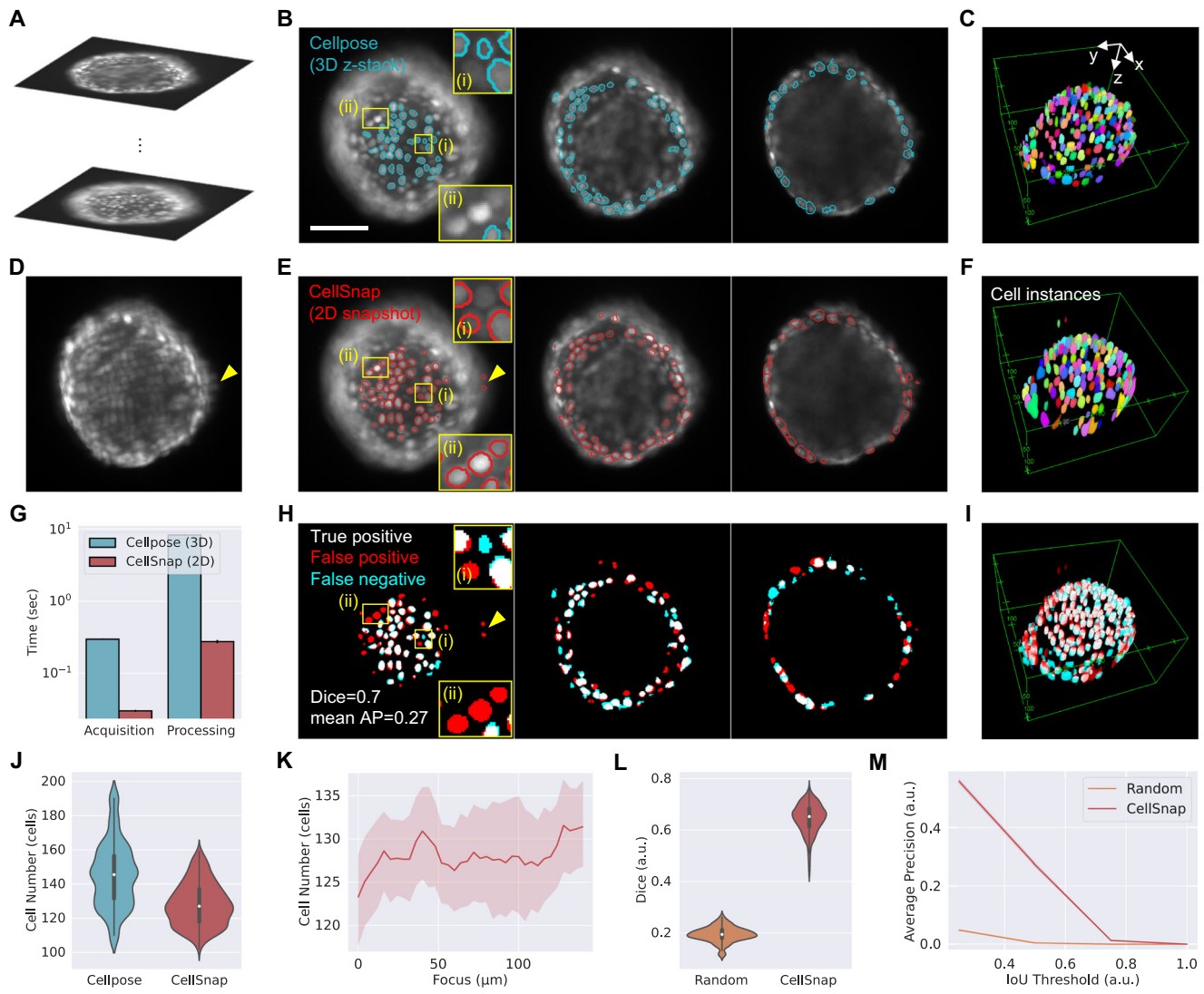

**Fig. 5 | CellSnap visual and quantitative results. A** Standard PSF z-stack. **B** Cellpose segmentations overlaid on top of raw axial slices separated by 40 μm. Scale bar is 100 μm. **C** Cellpose individual cell instances rendered in 3D. **D** Tetrapod snapshot roughly in the middle of the axial range. Yellow triangle marks a detaching cell going out of the axial range. **E** CellSnap segmentations overlaid on top of raw axial slices separated by 40 μm. **F** CellSnap individual cell instances (after watershed post-processing) rendered in 3D. **G** Bar plot comparison of acquisition and post-processing speeds in seconds. **H** Binary overlap of Cellpose (cyan, **B**) and CellSnap (red, **E**) segmentations; matches (true positives) are plotted in white. Zoom ins (i), (ii) (yellow rectangles) highlight correct segmentations identified as false positives (red) either (i) due to imperfect labels or (ii) due to inaccurate axial position. **I** Binary overlap 3D rendering. **J** Number of recovered cells comparison. Boxplot parameters were minima = 110/102, maxima = 190/159, 25% quantile = 130/118, 75% quantile = 159.5/137, median = 145.5/127, and mean = 145.4/128 for Cellpose/CellSnap respectively. **K** Number of recovered cells as a function of snapshot focus. Data is presented as mean value ± 95% confidence interval (i.e., 1.96*SEM). **L** Dice coefficient comparison. Boxplot parameters were minima = 0.11/0.43, maxima = 0.27/0.76, 25% quantile = 0.18/0.61, 75% quantile = 0.21/0.68, median = 0.19/0.65, and mean = 0.19/0.64 for Random/CellSnap respectively. **M** Average precision as a function of the Intersection Over Union (IoU) threshold. Data is presented as a mean value ± 95% confidence interval (i.e., 1.96*SEM). Source data are provided as a Source Data file.

## Methods

### Phase-mask mounts

**Modality 1.** A thin aluminum annular mount was fabricated with CNC, having an M25×0.75 external thread corresponding to the microscope objective turret. Onto the mount, a phase mask was glued from its corners using UV-cured optical adhesive (NOA 68, Norland). The aperture of the mount, at a diameter of 12 mm, corresponds to the size of the objective BFP and the mask. The mount with the mask was then screwed into the turret below the objective.

**Modality 2.** A mask mount was attached to the bottom of the objective itself rather than the turret. This was done by exploiting an RMS-M25 threading adapter native to the objective. We added a ring above the threading adapter, resulting in it protruding 2 mm below the objective.

Having done so, a 3D printed mount could be fit into the protruding internal RMS threading. The mount contains a centered aperture and space for the mask to be glued onto it, coinciding with the aperture. The advantage of the second modality is that it is easier to insert and remove the mount and that the mount does not occupy threaded turret space.

### Optical system and photolithographic phase mask design

The optical system consists of the native imaging system of Incu-cyte® S3 Live-Cell Analysis System, comprising an objective and a tube lens of 104 mm focal length. In all experiments besides NTA we used the 10X objective, with an NA of 0.3, on which we mounted phase elements. For NTA, the 20X with NA of 0.45 was used.

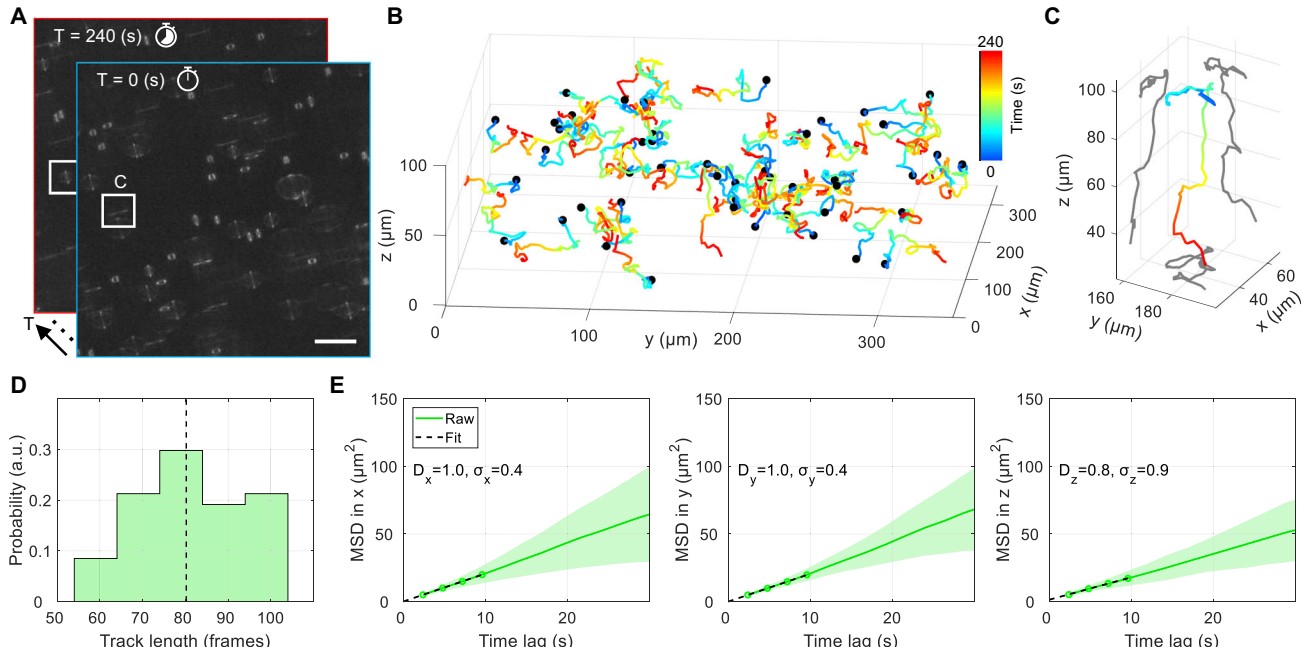

**Fig. 6 | Three-dimensional tracking with DeepSTORM3D. A** Time-lapse of snapshots with the Tetrapod PSF. Scale bar is 50 μm. **B** Recovered 3D tracks spanning 4 mins with a 2.4 s temporal resolution. **C** 3D track corresponding to the highlighted white square in (**A**). **D** Track length distribution (where 100 frames = 4 mins). Dashed black line marks the mean. **E** Ensemble mean square displacement (MSD) per axis. Filled area marks 1 standard deviation (SD). *D* and *s* are the per-axis estimated diffusion coefficient (μm²/s) and localization precision (μm), respectively. Source data are provided as a Source Data file.

## Tetrapod phase-mask optimization

To achieve precise 3D localization, we aim to minimize the Cramér–Rao lower bound (CRLB) as a proxy for the system's localization precision. The PSFs, once optimized through this process, facilitate precise 3D position estimation of individual emitters at position $\Theta = (x_0, y_0, z_0)$ based on noisy measurements (to simplify, we assume the system is governed by shot noise—for a complete model see ref. 24).

We denote $\mathrm{PSF}(x, y, M; \Theta)$ as the model PSF of an emitter at camera coordinates $(x, y)$, with a phase mask $M$. Assuming Poisson statistics, the measured image $I(x, y)$ is given by

$$I(x, y) \sim \mathrm{Poiss}(\mathrm{PSF}(x, y, M; \Theta) + B(x, y)). \tag{1}$$

where $B(x, y)$ is additive background noise, and $\mathrm{Poiss}(\cdot)$ is a Poisson noise operator corresponding to the shot-noise model. The log-likelihood function $L(\Theta)$ for the measurement in Eq. (1) is given by

$$L(\Theta) = \sum_{x, y}^{N_x, N_y} I(x, y) \cdot \log(\mathrm{PSF}(x, y, M; \Theta)) - \mathrm{PSF}(x, y, M; \Theta) + C(I(x, y)), \tag{2}$$

where $N_x, N_y$ are the number of image pixels per dimension, and $C(I(x, y))$ is a function of the measurements that is independent of $\Theta$.

Given the likelihood function $L(\Theta)$, the Fisher Information matrix $Q(\Theta)$ is defined as

$$[Q(\Theta)]_{i,j} = E\left[\left(\frac{\partial}{\partial \Theta_i} L(\Theta)\right) \cdot \left(\frac{\partial}{\partial \Theta_j} L(\Theta)\right) | \Theta\right]. \tag{3}$$

Substituting the log-likelihood from Eq. (2), we get

$$[Q(\Theta; M)]_{i,j} = \sum_{x, y}^{N_x, N_y} \frac{\partial}{\partial \Theta_i} \mathrm{PSF}(x, y, M; \Theta) \cdot \frac{\partial}{\partial \Theta_j} \mathrm{PSF}(x, y, M; \Theta)$$
$$\cdot \frac{1}{\mathrm{PSF}(x, y, M; \Theta) + B(x, y)}, \tag{4}$$

And the CRLB for estimating $\Theta_i \in (x_0, y_0, z_0)$ is defined as

$$\mathrm{CRLB}_i(\Theta; M) = [Q(\Theta; M)^{-1}]_{i,i} \tag{5}$$

In our implementation, $\mathrm{CRLB}_i(\Theta; M)$ is sampled at 100 equal intervals in the axial range, giving our optimization cost function:

$$\mathrm{Cost}(M) = \Sigma_z \Sigma_i \mathrm{CRLB}_i(x_0 = 0, y_0 = 0, z; M) \tag{6}$$

We also simplify the per-pixel background term $B(x, y)$ to a single scalar and scale the PSFs to match realistic signal counts encountered in SMLM imaging. Notably, different from the original proposition of a CRLB-optimized phase mask[15], we optimized the CRLB using a per-pixel approach rather than constraining the solution to a subspace of Zernike polynomials, as has been previously suggested[30]. This was particularly important to navigate the wide variety of possible solutions.

## EDOF phase-mask optimization

The PSF simulation model in the EDOF design was similar to the description above, although for EDOF, the cost function was different. Here, instead of minimizing the CRLB of 3D localization precision, the cost function used was the mean square difference of the PSF from a narrow 2D Gaussian profile. Here, a non-changing Gaussian profile was set as the target PSF (the algorithm is relatively robust to the exact size of this Gaussian), at equal axial intervals spanning the 60 μm axial design range.

## Photolithographic phase mask fabrication

The optimized phase profiles were converted to depth maps to be etched in quartz, considering the emission wavelength of the design (in our work, each mask corresponds to one of the two emission channels available in the HTP microscope). Using the depth maps, the phase masks were etched using photolithography (*Holo/Or*, Israel).

## Incoherence-based EDOF masks

The optical element comprised of stacked 0.17 mm-thick microscope coverslips (Merienfeld Cat. # 0107052), with holes cut using a UV laser cutter. The coverslip thickness provided sufficient optical path difference per layer, i.e., with the refractive-index difference between the element material and air of $\Delta n \approx 0.5$, the path difference is about 85 μm, while the typical fluorescence coherence length is on the order of 10 μm. We measured the fluorescence coherence length using an interferometer, and the results are consistent with emission spectrum-based calculation (Supplementary Note 2). The element comprises of 3 glass layers, resulting in 4 non-interfering (mutually incoherent) zones in the BFP. The proper diameters were calculated[24,26] such that each coherent zone produces the same DOF—given an objective aperture of 12 mm, the diameters of the glass layers are [6, 8.5, 10.4] mm. We used a laser-cutting machine (Epilog Zing 16) to cut holes in the coverslips and an outer frame to maintain the concentricity of the holes. After the coverslips were cut and cleaned, they were stacked inside a custom-designed 3D-printed mount that attaches to the bottom of the objective lens (Fig. 1C).

## Spheroid preparation

FaDu cells (courtesy of the Shamay lab, Technion) were grown in DMEM high glucose w/o L-glutamine and w/o sodium pyruvate (Sartorius, Israel), supplemented with 0.02% fetal bovine serum (Biological Industries, Israel), 0.01% L-glutamine and 0.01% penicillin-streptomycin solution (Biological Industries, Israel), in 25 cm² flask at 37 °C, 5% $CO_2$, until full coverage.

For spheroid formation, the cells were detached using 1 ml Trypsin-EDTA solution, transferred into a 15 ml falcon with 9 ml fresh medium, and centrifuged for 2 min, 893×$g$. The supernatant was removed, and the cells were resuspended in 10 ml fresh medium. The cell solution was diluted to 15,000 cell/ml, transferred into Corning® Costar® Ultra-Low Attachment 96-well plate (Merck), 200 μl per well (3000 cells), and incubated at 37 °C, 5% $CO_2$ for 4–7 days until spheroid formation.

For live cell nucleus labeling and imaging, each spheroid was stained by adding 1 μl of 0.15 mM SYTO®24 Green Fluorescent Nucleic Acid Stain (diluted with medium) (Invitrogen) into each well and incubated for 1 hour on a rocker at RT.

## Bead sample preparation and imaging

We used Mini-PROTEAN gel casting stand (BioRad) with 0.75 mm short glass to prepare acrylamide gel with beads by vertical solidification to allow beads to sparsely spread out in 3 dimensions. Gel solution with beads was prepared by gently mixing 1.2 ml double distilled water, 750 μl 40% acrylamide/bis-acrylamide solution (Sigma, A7802), 80 μl 50% glycerol, 3 μl bead solution (1 μm, fluoro-max dyed green aqueous fluorescent particles (Thermo Fisher Scientific, G0100)), 1.5 μl TEMED, and 10 μl 10% ammonium per sulfate. The gel solution was immediately pipetted into the gel casting stand and gently topped with about 1 ml of double-distilled water to level the gel. Once solidified, the gel was kept in water in the dark at 4 °C. Imaging comprised of a z-stack spanning 300 μm, at 1 μm step. Each FOV was imaged once with the standard PSF and once with EDOF, realized by a lithography-fabricated phase mask. Exposure time was 1 ms per frame for both PSFs.

## Nanoparticle tracking analysis sample preparation

Fluorescent microspheres (FF806, Invitrogen) were diluted in DDW to a concentration of $10^{-6}$. A mixture of 20% glycerol (by volume) and water was prepared. The bead solution was further diluted with 20% glycerol/water(v/v) to $10^{-7}$. To avoid sample evaporation we used 25 μl Gene Frame sticker (Thermo Fisher, AB0576) on a coverslip, added 25 μl bead solution and sealed the sample with another coverslip.

## Imaging—General

Imaging was performed using customized acquisition software on the Incucyte® S3 Live-Cell Analysis System equipped with a standard Green/Red fluorescence module (Green Excitation 440–480 nm, Green Emission 504–544 nm; Red Excitation 565–605 nm, Red Emission 625–705 nm). All references to the "standard PSF" address the native PSF of the Incucyte microscope. Comparisons between engineered and standard PSFs were obtained with similar objectives (either ×10 or ×20 original Incucyte objective) in all experiments.

## Spheroid imaging—EDOF

Each well of a 96-well plate was prepared with a single spheroid inside (see "spheroid preparation"). Imaging comprised of two z-stacks per well—one with the standard PSF for ground-truth data, followed by an objective comprising an EDOF phase element (layered glasses - incoherence-based EDOF). Z-stacks comprised 60 frames with axial steps of 5 μm. Exposure times were 0.49 ms for the standard PSF and 1.22 ms for the EDOF PSF, determined by a 2× base exposure for the EDOF as well as an additional 23% exposure time to compensate for signal reduction from photobleaching per z-stack.

## Spheroid imaging—Tetrapod

Each well of a 96-well plate was prepared with a single spheroid inside (see "spheroid preparation"). Imaging comprised of two z-stacks per well—one with the standard PSF for ground-truth data, and one with the Tetrapod PSF. Z-stacks comprised 100 frames with axial steps of 4 μm. Each well plate was scanned 3 times, where it was gently shaken between scans. By shaking the well plate, the spheroids had changed their positioning, providing a random new view, and we regarded them as new spheroids altogether. In total, data from four different plates was used. Per scan, fluorescence intensity dropped by 40% due to photobleaching, thus we increased exposure by the same factor with every repeated scan of a spheroid. Base exposure for the unmodified objective was 0.5 and 1.5 ms for the Tetrapod objective.

## Nanoparticle tracking analysis imaging

Imaging was conducted using Incucyte 20X objective, and a Tetrapod phase mask designed for the objective and the red emission channel. Imaging was performed with an exposure time of 400 and 2000 ms delay between images.

## Statistics and reproducibility

Experimental z-stacks for $x$–$z$ projections (Figs. 2A, 3B) were acquired once using a bright, isolated fluorescent bead on a coverslip. The same bead was used for both standard and EDOF data.

Data of dense beads in gel (Fig. 2) was acquired using a single sample. Five different z-stacks were acquired (Different FOVs). We chose one FOV where we visually observed an ROI that showcases multiple beads that are laterally close but axially apart, evidently showing the benefit of EDOF (Displayed in Fig. 2D). Only that FOV was analyzed using ThunderStorm, and statistics were done over 623k (1232k) localizations of $\sigma < 5$ μm of the standard (EDOF) PSF.

Spheroid EDOF data (Fig. 3C) was acquired over 96 spheroids in a 96-well plate. A single spheroid was chosen to visually showcase the advantage of our method. No statistical or quantitative analysis was performed on this data.

Quantitative analysis of the EDOF performance was done using a single z-stack acquisition of dense beads in gel (Fig. 2), comprising of thousands of localizations, as specified in the text.

Sample size in 3D reconstruction experiments with CellSnap (Figs. 4 and 5J–M) consisted of $n = 20$ cellular spheroids constructed from three biologically independent samples examined over four independent experiments with a 96-well plate. As detailed in Supplementary Note 6, our analysis excluded data from wells with saturated/

insufficient signals and where cells did not form spheroids. No statistical method was used to predetermine the sample size.

Nanoparticle tracking data was acquired for two different diffusing bead samples (Fig. 6). The same sample was imaged twice, once with the Tetrapod PSF and once with a standard imaging objective. The FOV, in Fig. 6, was chosen to highlight the enhanced-depth capabilities afforded by the Tetrapod PSF. Similar results were obtained for another four nonoverlapping FOVs from the same sample. Additionally, in Supplementary Fig. 13, we include the results of a similar experiment with a denser bead sample to highlight the applicability of our technique at higher emitter densities.

### Reporting summary

Further information on research design is available in the Nature Portfolio Reporting Summary linked to this article.

## Data availability

The data supporting the findings of this study is publicly available at Zenodo under accession code 10928122[50]: https://zenodo.org/records/10928122. Source data are provided with this paper.

## Code availability

Code is made publicly available at https://github.com/EliasNehme/HTPmicroscopy. The code release used to produce the results in this study is also publicly available at Zenodo under accession code 10938035[51]: https://zenodo.org/records/10938035.

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

## Acknowledgements

We'd like to thank Dana Meron Azagury from Yosef Shamay's lab for assisting with spheroid preparation and Amit Meller's lab for using their laser cutting machine (Zing 16, Epilog). This research was supported in part by Sartorius Stedim Biotech: N.O., N.Z., R.K.O.; by the Israel Innovation Authority, Technological Infrastructure Division, Applied Research in Academia; by the ISRAEL SCIENCE FOUNDATION (grant No. 450/18); by funding from the European Union's Horizon 2020 research and innovation program under grant agreement No. 802567-ERC-Five-Dimensional Localization Microscopy for Sub-Cellular Dynamics: Y.S., N.O., O.A., B.F., E.N., I.B.; Additionally, Y.S. is supported by the Zuckerman Foundation and by the Donald D. Harrington fellowship.

## Author contributions

N.O., R.K.O., B.F., and Y.S. conceived the experimental setup; E.N. conceived, designed, and implemented CellSnap; N.O., R.K.O., and N.Z. performed the experiments with technical assistance by P.K.; N.Z., I.B., and O.A. prepared the biological samples; B.F. designed the phase masks with contributions from E.N.; N.Z. ran deconvolution based post-processing over EDOF samples; E.N. analyzed the three-dimensional nanoparticle tracking data; N.O., E.N., and Y.S. did the writing.

## Competing interests

Y.S., B.F., N.O, E.N, and R.K.O. have published a patent (WO2022259243A1) relating to the wavefront shaping aspect described in this work. P.K. is employed by Sartorius Stedim North America, Inc. The remaining authors declare no competing interests.
