## [Peer Review File · Nature Communications]

Reviewers' Comments:

Reviewer #1:

Remarks to the Author:

In this paper, the authors have demonstrated an approach regarding how to realize the high throughput microscopy and 3D snapshot microscopy, where both of them are benefited from the PSF engineering by introducing additional phase-mask in the microscopy system. However, frankly speaking, I somewhat worry about that whether the novelty in this paper deserve to be published on Nature Communications. Because the way of PSF engineering used here to realize extended depth of field imaging (e.g. <https://doi.org/10.1145/3197517.3201333>; <https://doi.org/10.1364/OPTICA.1.000209>; <https://doi.org/10.1021/acsp Photonics.9b01703>; <https://doi.org/10.1109/TIP.2021.3060166>; <https://doi.org/10.1117/12.909273>;) and/or 3D/depth-imaging (<https://doi.org/10.1021/acsp Photonics.0c00354>;) has been proposed for a long time in the diffraction optics . For example, the first example in the paper of a high throughput microscopy is actually an extended depth of field imaging task by designing a long focus spot; the second example is actually 3D imaging by using a particular depth-sensitive PSF which can be related and varied with the object distance, such as Tetrapod PSF or double-helix PSF. I think that the real value for this paper is in their application and experiment that involves a crossover between diffraction optics and bioimaging, maybe along with an another small novelty to introduce the deep learning to build 3D results and achieve a snapshot effect avoiding a large time cost. But I think this is still inadequate to be published on Nature Communications, unless the authors can reconsidering their novelty and then persuade me again.

Besides, there are some critical issues in the paper that are unclear and the authors should clarify them as follows:

1. Authors should supply the phase-mask design information for both two examples in the paper respectively, including the phase profile and design process.
2. Please explain what Figure 2c means and the information that it reflects.
3. Please the authors answer in the paper that what the advantages of PSF engineering are for 3D imaging compared to other methods such as light-field imaging, lensless imaging and so on.
4. The authors say, they use deep learning to reconstruct the 3D imaging whose advantages are fast and small time costing. However, during the training process of the network, the authors should pre-measure lots of experimental data and images for both standard PSF and the Tetrapod PSF and at different depth. I think, this process is very time-consuming. What do authors think about this issue? If authors hope to discuss and compare the speed and time, the time for preceding training time should also be considered which is more fair in this way.
5. The authors say, "CellSnap was able to roughly match the performance of Cellpose". However, from Figure 5, comparing Fig.5c and Fig.5f, or looking at Fig. 5h, I think the difference between Cellsnap and Cellpose are very apparent. How to explain for this issue? If Fig.5c and Fig.5f are results with a same z-depth or not?

Reviewer #2:

Remarks to the Author:

Opatovski et al. have reported two methods for PSF engineering, one focuses on physical improvements, while the other combines deep learning-based algorithms. The authors demonstrated that this new methodology enables more efficient and accurate high throughput microscopy for cellular studies. However, the structure of this manuscript is loose, and therefore it is confusing and difficult to follow. The authors combined two different methods under a single topic, but they used different engineering approaches, validation, and biological applications. For each method, the authors had better introduce more details for quantitative analysis, instead of qualitative descriptions. While the results are intriguing and the integration of PSF engineering with deep learning-based algorithms is popular, the overall contribution to the field is incremental. Therefore, more substantial experimental results are required. In the meanwhile, the lengthy introduction to deep learning approaches distracts readers from the major focus of this manuscript. Overall, substantial revision is required to elucidate the technical novelty and biological applications of this manuscript before accessing its significance.

Major Comments:

- The paper lacks specific details about the design of the masks used in the two PSF methods and references other works for the design process. It is unclear whether the authors designed the masks themselves or followed existing approaches. The level of their contribution to the mask design process and any modifications or improvements made remains uncertain. Further information or clarification from the authors is needed to determine their involvement and unique contributions in mask design in at least abstract or result.
- The details of the deployment of phase masks are missing, but this is critical to assess the effectiveness of PSF modulation. Even though the authors provided the estimation of optical path differences, it still causes ambiguity regarding the differences in methods and positions used. More simulation or experimental results are needed for further evaluation. Additionally, it will make it easier for others who are not familiar with this direction to follow.
- Provide more information on the design and fabrication of the phase masks used to demonstrate the EDOF PSF and Tetrapod PSF. Specifically, discuss the specific materials or techniques employed and how they are optimized to achieve the desired PSF characteristics.
- Other information to check the accuracy of the model and the validity of data is needed. It is not mentioned what kind of optical system with what configuration the authors used to compare the results with the proposed masks and deep learning method. What system did the authors use in the quantification of DOF increase of incoherence EDOF to create the standard PSF?
- The paper mentions the possibility of using different phase masks for various applications, such as spectral or polarization modulation. Could the authors elaborate on the potential applications and benefits of using different types of phase masks in combination with compact PSF engineering?
- How could this method differentiate overlapping cells based on the modulated PSF in the 3D model? If the PSF is symmetrical along a specific direction, the spatial variation is difficult to be extracted along this direction.
- The authors should address potential trade-offs or practical constraints when implementing compact PSF engineering in high throughput microscopy, as well as the post-processing procedures.

Minor Comments:

- In Figure S2, B and D are not clear. Yellow arrows like Figure A and C are helpful. In C and D, "the ink spot placed ~20mm below the correct position" does not make sense. In F, clarification is needed for the reason why "element at wrong plane" at 0.75 is 7 and figure does not show any element at BFP.
- More details of the objective lens are encouraged to disclose. For example, NA, magnification, etc.
- Scale bars are fundamental in microscopy images. Unfortunately, some figures such as S2 and S4 do not include it. Figure legends and units are also needed. FWHM should be measured for each figure, for example, figure 2.
- The references to supplementary material (e.g., Supplementary Section 1) should be properly formatted and numbered for easy identification. ("For architecture details and learning hyper-parameters see supplementary sections neural network architectures and training details.")
- The authors mentioned post-processing using watershed splitting. Could they provide more details on how this step is applied in CellSnap and its role in improving the segmentation results?

Reviewer #3:

Remarks to the Author:

After carefully reading the manuscript #431629: "Depth-enhanced high throughput microscopy by compact PSF engineering" I consider that it can be published in nature communications provided a revision is performed.

The manuscript deals with the use of point spread function engineering in compact modality which allows high throughput (HTP) screening of 3d cell-cultures under incubation conditions. The phase-modulating mask is placed in the BFP of an objective lens and allows Depth of Field Extension as well as snapshot 3D cell segmentation from of Multi-Cellular Tumor Spheroids (MCTS)

when combined with Deep Learning techniques.

The experimental results support the proposal to a great extent. Besides, the manuscript is clearly written. However, I would like the authors to address the following issues before publication of the manuscript:

1) The state of the art in 3d visualization of MCTS, either by means of multifocus sensing or light sheet microscopy is not clearly stated:

I. Morphological information from multifocus fluorescence microscopy of MCTS has been addressed by means of Fourier domain post-processing, see for example:

Alonso, J. R., Silva, A., Fernández, A., & Arocena, M. (2022). Computational multifocus fluorescence microscopy for three-dimensional visualization of multicellular tumor spheroids. *Journal of Biomedical Optics*, 27(6), 066501-066501.

II. Large scale screening (>600 samples) of MCTS has been addressed by means of light-sheet fluorescence microscopy (LSFM) and with high throughput (120 samples /min), see for example:

Zhu, T., Nie, J., Yu, T., Zhu, D., Huang, Y., Chen, Z., ... & Fei, P. (2023). Large-scale high-throughput 3D culture, imaging, and analysis of cell spheroids using microchip-enhanced light-sheet microscopy. *Biomedical Optics Express*, 14(4), 1659-1669.

III. Also by means of light-sheet microscopy but performed live, cell-dynamics in MCTS has been addressed, see for example:

Desmaison, A., Guillaume, L., Triclin, S., Weiss, P., Ducommun, B., & Lobjois, V. (2018). Impact of physical confinement on nuclei geometry and cell division dynamics in 3D spheroids. *Scientific reports*, 8(1), 8785.

2) Although the PSF engineering approach can indeed give results in a snapshot (for example EDoF), the applications presented deal with static processes in MCTS, in comparison to dynamical processes as addressed in the previous reference. In this regard, acquisition or post-processing time in LSFM as well as fluorescence multifocus sensing would not represent a comparable disadvantage. I would like the authors to discuss on this issue.

3) Fig 1: a phase profile in radians of the mask placed at the BFP might be useful.

4) The effect of the phase mask on the axial imaging capabilities of their system is clearly stated. However, I would like the authors to comment on the possible effects of their PSF engineering approach on the lateral resolution.

Depth-enhanced high throughput microscopy by compact PSF engineering

Author Response to Referees

We wish to thank the editor and reviewers for the careful attention paid to our manuscript. Reviewer comments are shown in black; our response is in blue. Extracts from the manuscript are in italics.

In the revised manuscript, added/changed text red colored.

Reviewer #1:

In this paper, the authors have demonstrated an approach regarding how to realize the high throughput microscopy and 3D snapshot microscopy, where both of them are benefited from the PSF engineering by introducing additional phase-mask in the microscopy system. However, frankly speaking, I somewhat worry about that whether the novelty in this paper deserve to be published on Nature Communications. Because the way of PSF engineering used here to realize extended depth of field imaging (e.g. <https://doi.org/10.1145/3197517.3201333> [1]; <https://doi.org/10.1364/OPTICA.1.000209> [2]; <https://doi.org/10.1021/acsp Photonics.9b01703> [3]; 10.1109/TIP.2021.3060166 [4]; <https://doi.org/10.1117/12.909273> [5];) and/or 3D/depth-imaging (<https://doi.org/10.1021/acsp Photonics.0c00354> [6];) has been proposed for a long time in the diffraction optics . For example, the first example in the paper of a high throughput microscopy is actually an extended depth of field imaging task by designing a long focus spot; the second example is actually 3D imaging by using a particular depth-sensitive PSF which can be related and varied with the object distance, such as Tetrapod PSF or double-helix PSF.

I think that the real value for this paper is in their application and experiment that involves a crossover between diffraction optics and bioimaging, maybe along with an another small novelty to introduce the deep learning to build 3D results and achieve a snapshot effect avoiding a large time cost. But I think this is still inadequate to be published on Nature Communications, unless the authors can reconsidering their novelty and then persuade me again.

References:

- [1] End-to-end Optimization of Optics and Image Processing for Achromatic Extended Depth of Field and Super-resolution Imaging; GORDON WETZSTEIN
- [2] Extended depth-of-field imaging and ranging in a snapshot; PAUL ZAMMIT et al
- [3] Inverse Designed Metalenses with Extended Depth of Focus; Elyas Bayati
- [4] Learning Wavefront Coding for Extended Depth of Field Imaging; Ugur Akpınar et al
- [5] Simultaneous quantitative depth mapping and extended depth of field for 4D microscopy through PSF engineering; Ramzi et al.
- [6] Metasurface Generation of Paired Accelerating and Rotating Optical Beams for Passive Ranging and Scene Reconstruction; Shane Colburn and Arka Majumdar.

We appreciate the time and attention dedicated to our work. We have dug into the papers the reviewer referred us to, some were new to us and have been insightful. We are aware of the advancements and contribution of PSF engineering to various optical configurations. We wish to emphasize that the main novelty in our work, in that aspect, relates to the introduction and demonstration of PSF engineering to high-throughput-microscopy, i.e. when a cumbersome 4F setup is not applicable, opening up a variety of possibilities, rather than to a specific PSF design.

The advancement in research of optical/computational EDOF approaches is exactly the motivation to introduce this tool into HTP microscopy. HTP microscopy is a bioimaging domain where PSF engineering has not been implemented - we believe this is mostly due to practical reasons, that in this work we overcome.

As an additional use case, we added a nanoparticle tracking analysis (NTA) showing the benefit of snapshot 3D imaging in diffusion studies. This is yet another example, where there is a clear benefit for an extended z-range, yet volumetric NTA studies are rarely explored due to the difficulty of implementation with existing systems.

Besides, there are some critical issues in the paper that are unclear and the authors should clarify them as follows:

1. Authors should supply the phase-mask design information for both two examples in the paper respectively, including the phase profile and design process.

Thank you for the suggestion. We added the description in the following methods sections:

Tetrapod phase-mask optimization

EDOF phase-mask optimization

Photolithographic phase masks fabrication

Incoherence-based EDOF masks

Furthermore, phase profiles of the masks we used can be found in Figure 3.A and 4.A. We also added in the SI a table summarizing all mask types used in this work.

2. Please explain what Figure 2c means and the information that it reflects.

Figure 2.C presents a histogram of the localization widths found, when applying ThunderSTORM localization over the entire z-stack. The main conclusions from this histogram overlay is that many more PSFs were found by ThunderSTORM with EDOF, with a smaller mean size, as we would expect. These conclusions are summarized quantitatively in the main text:

“As result, twice as many localizations were obtained with the EDOF PSF, with an average width (standard deviation of the gaussian fit) of 1.39 μm . This is compared to 1.74 μm for the standard PSF (Fig. 2.C).”

An additional clarification to the figure was added in the main text:

“We evaluated the EDOF contribution by finding the PSF width as a function of misfocus (Fig. 2.B), as well as counting the number of localizations per width, presented in the histograms in Fig. 2.C.”

Also, the part regarding Fig 2.D was moved to the end of the paragraph for better readability.

3. Please the authors answer in the paper that what the advantages of PSF engineering are

We thank the reviewer for noticing this gap in our introductory sentence. The main reason PSF engineering is preferred over lenseless/light-field imaging is the lateral resolution that is usually worse with these techniques. In PSF engineering, this tradeoff can be optimized as desired for a given application, using for example PSF design methods such as Cramer Rao Lower Bound optimization, similar to the methodology explained in the new methods section **“Tetrapod phase-mask optimization”**. To emphasize this fact, we added the following sentence in the main text:

“Compared with PSF engineering, lensless and light-field imaging typically entail a degradation in lateral resolution, especially mask-only systems (such as a diffuser and a bare sensor) which have no magnifying optics and are thus limited to low effective numerical apertures (NA)”.

Additionally, please note that light-field imaging is a special case of PSF engineering and can be in principle achieved with our compact PSF engineering approach by employing a phase mask consisting of a set of multifocal microlenses mimicking the operation of a microlens array at the image plane. Hence, if this technique is desired for a given application, then our system can be seamlessly instantiated to achieve this modality (e.g., see ref [33] **miniscope** in the main text).

4. The authors say, they use deep learning to reconstruct the 3D imaging whose advantages are fast and small time costing. However, during the training process of the network, the authors should pre-measure lots of experimental data and images for both standard PSF and the Tetrapod PSF and at different depth. I think, this process is very time-consuming. What do authors think about this issue? If authors hope to discuss and compare the speed and time, the time for preceding training time should also be considered which is more fair in this way.

We thank the reviewer for raising this point. First, please note that this is one of the strengths of HTP microscopy, enabling the acquisition of large datasets for training with relative ease compared to standard microscopes. Second and more importantly, it is important to note that acquiring the training data is a one-time effort. After this process is done once, the resulting model can be applied to thousands of spheroids very rapidly, cutting down acquisition and post-processing times by an order of magnitude. In our experiments, we observed the resulting model to be robust. For example, we were able to apply the trained net to plates acquired on different days, with slightly different alignments of the optics, and some changes in the PSF. The model was still accurate and recovered the underlying cells faithfully. Hence, in cases where the experiment is repeated a large

number of times for statistical reasons, it is worth investing in acquiring a training set once, to enjoy the afforded speed benefits for many more samples later.

To make the reader alert of this point we added the following sentence to supplementary note 13 (**Acquisition and post-processing speeds**):

“Please note that in this analysis we left out the time required for training set acquisition and model training because these are one-time efforts that need to be done only once. The resulting model can then be applied to thousands of samples seamlessly, enjoying the speed up benefits mentioned above.”

5. The authors say, “CellSnap was able to roughly match the performance of Cellpose”. However, from Figure 5, comparing Fig.5c and Fig.5f, or looking at Fig. 5h, I think the difference between Cellsnap and Cellpose are very apparent. How to explain for this issue? If Fig.5c and Fig.5f are results with a same z-depth or not?

The reason for the apparent inaccuracies is a mixture of multiple error sources such as imperfect “ground truth” derived with Cellpose, sample movement in between the Standard PSF and the Tetrapod PSF acquisitions, and possible reconstruction shifts in z. We thank the reviewer for pointing out this confusion. To address it at length, we added a supplementary note (**Imperfect labels and axial reconstruction accuracy**) referred to in the main text, and introduced two supplementary figures with visual examples to illustrate our explanations (Fig.S8 and Fig.S9). In short, some of the cells that appear as reconstruction mistakes, are in fact correct and classified as errors only due to sample movement or imperfect ground truth derived automatically with Cellpose. In addition, some reconstructed cells are slightly shifted in z (4 μm shifts compared to a 20 μm cell span in z), and still appear wrong because of the chosen z-slice presenting the 3D reconstruction only at a single given plane.

Reviewer #2 (Remarks to the Author)

Opatovski et al. have reported two methods for PSF engineering, one focuses on physical improvements, while the other combines deep learning-based algorithms. The authors demonstrated that this new methodology enables more efficient and accurate high throughput microscopy for cellular studies. However, the structure of this manuscript is loose, and therefore it is confusing and difficult to follow. The authors combined two different methods under a single topic, but they used different engineering approaches, validation, and biological applications. For each method, the authors had better introduce more details for quantitative analysis, instead of qualitative descriptions. While the results are intriguing and the integration of PSF engineering with deep learning-based algorithms is popular, the overall contribution to the field is incremental. Therefore, more substantial experimental results are required. In the meanwhile, the lengthy introduction to deep learning approaches distracts readers from the major focus of this manuscript. Overall, substantial revision is required to elucidate the technical novelty and biological applications of this manuscript before accessing its significance.

The introduction could be reorganized. A layout of the paper may focus the reader on the agenda, which is the placement of a phase element that we later justify with various examples.

We thank the reviewer for bringing the matters to our attention. Alongside the point-by-point adjustments to the paper that will be specified ahead, we have made the introductory section more concise by changing the last paragraph. The new version, we hope, will help clarify the aim of the paper and help with the general organization of the manuscript.

We replaced:

In this study, two PSFs are demonstrated – an EDOF PSF(Abrahamsson et al., 2006; Nehme et al., 2021) and the Tetrapod PSF(Shechtman et al., 2014a). The EDOF PSF maximizes the intensity at the PSF center over an extended axial range, far beyond that of the standard PSF, while the Tetrapod PSF is optimal for 3D localization precision of individual emitters over a given axial range. The goal of the EDOF PSF is to provide an extended acquisition depth without the need for post-processing, i.e., to enable a single-shot (or few-shot) z-projection. On the other hand, the Tetrapod PSF provides 3D information over an extended range via post-processing.

With:

This work presents our implementation of compact PSF engineering in HTP. We showcase two cornerstone PSFs from the field of PSF engineering - an EDOF PSF(Abrahamsson et al., 2006; Nehme et al., 2021) and the Tetrapod PSF(Shechtman et al., 2014a). The EDOF PSF maximizes the on-axis intensity over an extended axial range, to provide an extended acquisition depth without the need for post-processing. The Tetrapod PSF is optimal for 3D localization of individual emitters over a given axial range, thus providing 3D information via post-processing. We describe and analyze both PSFs experimentally, discuss the tradeoffs they exhibit, and through them, shed light on the opportunities in applying PSF engineering to HTP.

Major Comments:

- The paper lacks specific details about the design of the masks used in the two PSF methods and references other works for the design process. It is unclear whether the authors designed the masks themselves or followed existing approaches. The level of their contribution to the mask design process and any modifications or improvements made remains uncertain. Further information or clarification from the authors is needed to determine their involvement and unique contributions in mask design in at least abstract or result.

We thank the reviewer for this comment, which prompted us to clarify issues involving the phase masks used. The masks were fully designed by the authors, specifically for the applications presented in the paper. Because the focus of the paper is the implementation and use of the masks, rather than the design of the masks, we chose to refrain from going into too much detail regarding the masks in the abstract or the results sections. Nevertheless, the design and fabrication of the masks is an important part of the work, and we are glad this was brought to our attention. We now fully describe these processes in the following methods sections:

Optical system and photolithographic phase masks design

Tetrapod phase-mask optimization

EDOF phase-mask optimization

Photolithographic phase masks fabrication

Incoherence-based EDOF masks

- The details of the deployment of phase masks are missing, but this is critical to assess the effectiveness of PSF modulation. Even though the authors provided the estimation of optical path differences, it still causes ambiguity regarding the differences in methods and positions used. More simulation or experimental results are needed for further evaluation. Additionally, it will make it easier for others who are not familiar with this direction to follow.

We added detailed description of how the masks were designed in the methods sections:

Optical system and photolithographic phase masks design

Tetrapod phase-mask optimization

EDOF phase-mask optimization

Photolithographic phase masks fabrication

Incoherence-based EDOF masks

In addition, we added a supplementary table that summarizes all the phase masks used in the experiments.

We hope that this clarifies the deployment of the phase masks across the different sections of the manuscript. Together with the careful validation of the PSF quality by comparison with simulation and ground-truth, the effectiveness of PSF modulation is clearly analyzed.

- Provide more information on the design and fabrication of the phase masks used to demonstrate the EDOF PSF and Tetrapod PSF. Specifically, discuss the specific materials or techniques employed and how they are optimized to achieve the desired PSF characteristics.

This is now addressed in the revised methods sections:

Tetrapod phase-mask optimization

EDOF phase-mask optimization

Photolithographic phase masks fabrication (this addresses both Tetrapod and EDOF masks)

Incoherence-based EDOF masks

- Other information to check the accuracy of the model and the validity of data is needed. It is not mentioned what kind of optical system with what configuration the authors used to compare the results with the proposed masks and deep learning method. What system did the authors use in the quantification of DOF increase of incoherence EDOF to create the standard PSF?

In all methods, the comparisons made were relative to an equivalent unmodified imaging system, namely, the same objective (native Incucyte objective) without a phase mask in the optical path. This is what we referred to as the “standard PSF”.

We have added a clarification in “Methods”, under “Imaging – general”.

Regarding the accuracy of the model, this was the main drive for performing experimental tests for the PSFs we described in the paper. For EDOF, we have compared the simulated and experimental results (Figure 2.A), and tested the increase in DOF under objective conditions (fitting statistics using a common fitting tool – ThunderSTORM). We figured this approach is more sensible than direct PSF evaluation and comparison, as often DOF quantification in EDOF PSFs can vary significantly depending on the metrics used, thus may be misleading. Validity of the Tetrapod-spheroid results was done by comparing reconstructions to z-stack reconstructions with the standard PSF, obtained with the best 3D segmentation tool commonly available (CellPose). This is shown in the revised figures 5, S7-S11.

- The paper mentions the possibility of using different phase masks for various applications, such as spectral or polarization modulation. Could the authors elaborate on the potential applications and benefits of using different types of phase masks in combination with compact PSF engineering?

A phase mask can encode spectral information(Hershko et al., 2019), and even provide 3D information at the same time(Opatovski et al., 2021). For example, a multi-color optimized phase mask could facilitate simultaneous imaging of multiple colors on the same camera, which would increase imaging throughput in multi-label experiments. A mounted element can also have polarization properties, which may be of significance for biological imaging for example, polarized light microscopy (Oldenbourg, 2013).

We have added relevant references in the main text.

- How could this method differentiate overlapping cells based on the modulated PSF in the 3D model? If the PSF is symmetrical along a specific direction, the spatial variation is difficult to be extracted along this direction.

This is a very interesting question that warrants empirical validation. Following the reviewer’s remark, we examined our reconstructions to locate such cases in the data. Our findings were the following:

1. Such detrimental cell arrangement is a rare event given the finite cellular cytoplasm dimensions, and does not happen often in our case.
2. In the rare event of adjacent cells aligning exactly with the principal axis of the Tetrapod PSF, the reconstruction accuracy was indeed reduced, mainly due to inaccurate post-processing such that nearby reconstructed cells were difficult to separate.

In the revised manuscript we address this intricate point in supplementary note “**Post-processing segmentation outputs**” and Fig.S7.

- The authors should address potential trade-offs or practical constraints when implementing compact PSF engineering in high throughput microscopy, as well as the post-processing procedures.

In essence, every PSF is characterized by its distinctive trade-offs, which are the main reason for choosing/using an engineered PSF.

The general trade-offs in the context of PSF engineering are that on the plus side, the PSF is modified to encode additional information into the image. Be it 3D information, extended imaging depth, spectral information or more. The former two are the benefits we discussed in the manuscript. The downsides are the procedure of setting up the optical system to engineer the PSF, (often) the requirement for post processing and possibly a reduction in resolution. For example, the in-focus standard PSF produces the optimal lateral resolution for a single emitter, but this PSF is inferior for out-of-focus objects.

We have chosen not to dive into the general trade-offs in PSF engineering as a whole, as they are discussed extensively in the literature (Shechtman et al., 2014b; Von Diezmann et al., 2017) Instead, we have elaborated on the results we demonstrate – how the lateral resolution of the EDOF is reduced for the improvement in DOF, and how using CellSnap required training and post-processing steps, but it enables single-shot 3D segmentation that is robust to misfocus.

We have also discussed the implementation of our approach. While it's not as easy-to-implement as the standard PSF (which requires no work at all), it's an add-on that's simple to insert and remove in an instant, which is far easier and more compact compared to the common approach of PSF engineering that requires building a 4-f system.

We now added in the discussion:

Incorporating PSF engineering in an optical system to increase the information throughput is subject to trade-offs. Most notable is the engagement with the optics, which in our work is done with minimal effort as an easily applied/removed add-on. Other drawbacks may include the requirement for post-processing, and resolution compromise, as we have presented in this work.

Minor Comments:

- In Figure S2, B and D are not clear. Yellow arrows like Figure A and C are helpful. In C and D, "the ink spot placed ~20mm below the correct position" does not make sense. In F, clarification is needed for the reason why "element at wrong plane" at 0.75 is 7 and figure does not show any element at BFP.

We appreciate the attention paid to this section. Indeed, the section was not clear enough. We have substantially modified the text and the figure, providing a better description of the experiment. We added side-views of the setup and the ink-dotted glass - we hope this makes things much clearer, in a way that now makes sense. In addition, we explain the histogram in the main text. The quiver plots representing the drift of the dark spot relative to the PSF are now displayed in distinct gradient colors, which make it easier to understand (and see the radial pattern in S3.G). Also, as the reviewer noted, the 0.75 bar

seems odd, as both datasets display strange values inside that bar (0 and 7), albeit this is coincidental.

- More details of the objective lens are encouraged to disclose. For example, NA, magnification, etc.

Added in the section “Optical system and phase-mask design” in methods.

- Scale bars are fundamental in microscopy images. Unfortunately, some figures such as S2 and S4 do not include it. Figure legends and units are also needed. FWHM should be measured for each figure, for example, figure 2.

Thank you for bringing this to our attention. Scale bars' legend and units are verified in each figure.

In addition, the horizontal intensity distributions in of the PSFs in Fig. 2 were added below. Each has a gaussian fit for FWHM calculation (where applicable). We have chosen the use of σ instead of FWHM to describe the width of a gaussian fit as it is the output from ThunderSTORM. Nevertheless, based on the reviewer's suggestion, we have added FWHM values for Fig. 2, and reminded the relation between FWHM to σ , to simplify reading for readers who are used for the FWHM measure.

Horizontal cross-section gaussian fits were added to figure 2 (bottom), and FWHM values. The following was added to the caption:

“The bottom row shows intensity profiles of the horizontal lines, with a gaussian fit where applicable (the standard PSF profile of $z = 280, 290 \mu\text{m}$ is not gaussian). FWHM of the gaussian profiles is specified near each plot, where $\text{FWHM} \approx 2.36\sigma$. In the legend, “St.” stands for the standard PSF”

- The references to supplementary material (e.g., Supplementary Section 1) should be properly formatted and numbered for easy identification. (“For architecture details and learning hyper-parameters see supplementary sections neural network architectures and training details.”)

Thank you for the comment. SI sections are now numbered, and a table of contents was added.

- The authors mentioned post-processing using watershed splitting. Could they provide more details on how this step is applied in CellSnap and its role in improving the segmentation results?

Thanks for pointing out this missing detail. We added a supplementary note (“**Post-processing segmentation outputs**”) covering the role of post-processing, including

Fig.S6 for a visual before/after inspection. In short, the post-processing mainly separates touching cell instances to label them individually, while discarding cells with an oddly small number of voxels.

Reviewer #3 (Remarks to the Author)

After carefully reading the manuscript #431629: "Depth-enhanced high throughput microscopy by compact PSF engineering" I consider that it can be published in nature communications provided a revision is performed.

The manuscript deals with the use of point spread function engineering in compact modality which allows high through put (HTP) screening of 3d cell-cultures under incubation conditions. The phase-modulating mask is placed in the BFP of an objective lens and allows Depth of Field Extension as well as snapshot 3D cell segmentation from of Multi-Cellular Tumor Spheroids (MCTS) when combined with Deep Learning techniques.

The experimental results support the proposal to a great extent. Besides, the manuscript is clearly written. However, I would like the authors to address the following issues before publication of the manuscript:

1) The state of the art in 3d visualization of MCTS, either by means of multifocus sensing or light sheet microscopy is not clearly stated:

I. Morphological information from multifocus fluorescence microscopy of MCTS has been addressed by means of Fourier domain post-processing, see for example:

Alonso, J. R., Silva, A., Fernández, A., & Arocena, M. (2022). Computational multifocus fluorescence microscopy for three-dimensional visualization of multicellular tumor spheroids. *Journal of Biomedical Optics*, 27(6), 066501-066501.

II. Large scale screening (>600 samples) of MCTS has been addressed by means of light-sheet fluorescence microscopy (LSFM) and with high through put (120 samples /min), see for example:

Zhu, T., Nie, J., Yu, T., Zhu, D., Huang, Y., Chen, Z., ... & Fei, P. (2023). Large-scale high-throughput 3D culture, imaging, and analysis of cell spheroids using microchip-enhanced light-sheet microscopy. *Biomedical Optics Express*, 14(4), 1659-1669.

III. Also by means of light-sheet microscopy but performed live, cell-dynamics in MCTS has been addressed, see for example:

Desmaison, A., Guillaume, L., Triclin, S., Weiss, P., Ducommun, B., & Lobjois, V. (2018). Impact of physical confinement on nuclei geometry and cell division dynamics in 3D spheroids. *Scientific reports*, 8(1), 8785.

We thank you for the enlightening suggestions. These papers are indeed relevant to the field covered in this manuscript and the papers were cited accordingly:

References 1,2 have been added (here, [10] and [11]):

original text:

“While sophisticated optical solutions such as high throughput light sheet microscopy do improve some aspects [9], such as minimizing photodamage and reducing background, acquisition...”

Replaced with:

“While sophisticated optical solutions such as high throughput light sheet microscopy [9,10], or axial scanning using a focus-tunable lens [11] do provide axial information by axial scanning, acquisition...”

Ref 3 has been added in the following (here [9]):

- *“... solution for acquiring 3D information is to axially scan each sample at multiple focal multiple focal planes, by acquiring a z-stack[9]”*
- *“... any quantitative analysis performing phenotypic measurements on the single cell level[9]”*

2) Although the PSF engineering approach can indeed give results in a snapshot (for example EDoF), the applications presented deal with static processes in MCTS, in comparison to dynamical processes as addressed in the previous reference. In this regard, acquisition or post-processing time in LSFM as well as fluorescence multifocus sensing would not represent a comparable disadvantage. I would like the authors to discuss on this issue.

Thank you for the suggestion. Indeed, we have only presented static scenes, which do not emphasize the speed benefits of snapshot imaging enabled by PSF engineering. Following this comment, we sought a sample that would exhibit interesting dynamics. Thus, we have added a nanoparticle tracking analysis experiment as an example of a dynamic scene (Fig. 6 main text). Aside from acquisition speed, another benefit of snapshot imaging is the reduced exposure of the sample to light. This can be imperative in samples that suffer from photodamage or photobleaching. These are well-known challenges in fluorescence microscopy of biological samples.

3) Fig 1: a phase profile in radians of the mask placed at the BFP might be useful.

We appreciate the suggestion. We too believe it might be useful, also as an introduction to PSF engineering. We chose to refer here to a later figure with the exact phase pattern, for the following reason: in this early part of the paper we wish to emphasize the concept of the paper, which is the compact implementation of phase modulation. As one can see, we actually present multiple PSFs types in the figure, and the general concept of PSF modulation in the right side of the figure. We are worried that presenting a generic phase profile at this stage might deflect the reader from the main goal of the figure. Phase profiles

of the masks we used are presented in Figures 3 and 4, where they are specifically addressed.

4) The effect of the phase mask on the axial imaging capabilities of their system is clearly stated. However, I would like the authors to comment on the possible effects of their PSF engineering approach on the lateral resolution.

Naturally, in all PSF-engineering applications, the introduction of 3D information encoding by an engineered PSF comes at a cost in lateral resolution compared to the in-focus standard PSF. In general, it is difficult to quantitatively assess the effect of the PSF on the lateral resolution of the reconstruction of non-pointlike objects, as this property is dependent on the examined z-slice (specific PSF shape), the content of the reconstructed scene, and even the angle at which this property is measured (see for example new supplementary note “**Post-processing segmentation outputs**”). However, we added an NTA experiment following the reviewer’s previous point regarding scene dynamics. As a byproduct, we can assess the lateral resolution in this experiment by extrapolating a linear fit to the mean square displacement (MSD) plots at a time lag of 0 (Fig. 6 main text, supplementary note “Three-dimensional tracking”, and Figs.S12-S13). The result indicates that for single point sources our lateral localization precision with the Tetrapod PSF is $\sim 0.4\mu\text{m}$, compared to roughly $\sim 0.9\mu\text{m}$ in z.

References

- Abrahamsson, S., Usawa, S., & Gustafsson, M. (2006). *A new approach to extended focus for high-speed, high-resolution biological microscopy* (J.-A. Conchello, C. J. Cogswell, & T. Wilson, Eds.; p. 60900N). <https://doi.org/10.1117/12.647022>
- Hershko, E., Weiss, L. E., Michaeli, T., & Shechtman, Y. (2019). Multicolor localization microscopy and point-spread-function engineering by deep learning. *Optics Express*. <https://doi.org/10.1364/oe.27.006158>
- Nehme, E., Ferdman, B., Weiss, L. E., Naor, T., Freedman, D., Michaeli, T., & Shechtman, Y. (2021). Learning Optimal Wavefront Shaping for Multi-Channel Imaging. *IEEE Transactions on Pattern Analysis and Machine Intelligence*, 43(7), 2179–2192. <https://doi.org/10.1109/TPAMI.2021.3076873>
- Oldenbourg, R. (2013). Polarized Light Microscopy: Principles and Practice. *Cold Spring Harbor Protocols*, 2013(11), pdb.top078600. <https://doi.org/10.1101/pdb.top078600>
- Opatovski, N., Shalev Ezra, Y., Weiss, L. E., Ferdman, B., Orange-Kedem, R., & Shechtman, Y. (2021). Multiplexed PSF Engineering for Three-Dimensional Multicolor Particle Tracking. *Nano Letters*, 21(13), 5888–5895. <https://doi.org/10.1021/ACS.NANOLETT.1C02068>
- Shechtman, Y., Sahl, S. J., Backer, A. S., & Moerner, W. E. (2014a). Optimal Point Spread Function Design for 3D Imaging. *Physical Review Letters*, 113(13), 133902. <https://doi.org/10.1103/PhysRevLett.113.133902>

- Shechtman, Y., Sahl, S. J., Backer, A. S., & Moerner, W. E. (2014b). Optimal Point Spread Function Design for 3D Imaging. *Physical Review Letters*, 113(13), 133902.
<https://doi.org/10.1103/PhysRevLett.113.133902>
- Von Diezmann, A., Shechtman, Y., & Moerner, W. E. (2017). Three-Dimensional Localization of Single Molecules for Super-Resolution Imaging and Single-Particle Tracking. In *Chemical Reviews* (Vol. 117, Issue 11, pp. 7244–7275).
<https://doi.org/10.1021/acs.chemrev.6b00629>

Reviewers' Comments:

Reviewer #1:

Remarks to the Author:

The authors have solved all my questions and confusions. I consider it could be published in nature communications.

Reviewer #2:

Remarks to the Author:

The authors demonstrated adeptness in addressing the reviewer's concerns. The strategic relocation of supplementary explanations from the main paper to the supplementary information notably enhanced the article's clarity and accessibility. I commend the authors for taking this positive step. Furthermore, I would appreciate it if the authors could consider addressing the following concerns to enhance the overall quality of the manuscript further.

- Regarding the question concerning the advantages of the mentioned methods over the light field, the authors briefly highlighted the lower lateral resolution of light field. However, with the recent advancements in deep learning-based post-processing computations for light field reconstruction, researchers have successfully achieved significantly improved lateral resolution compared to previous methods. Notably, even in light field deconvolution, the number of iterations can impact output resolution. Furthermore, in the context of 3D data, axial resolution holds significance. Therefore, it is crucial to inform readers about the advantages or disadvantages of the discussed methods compared to light field techniques.
- The authors mentioned deconvolution's impact on output quality but did not specify the algorithm used. Clarifying the deconvolution algorithm and discussing the effects of iteration variation on final resolution would enhance transparency and reader understanding of the methodology.
- The authors have not made simulation codes available online, which could improve the reliability and understanding of their data. I recommend they consider sharing the codes online for transparency and reproducibility.

Reviewer #3:

Remarks to the Author:

The authors have carefully revised the paper based on the comments provided by the reviewers. In particular, new experiments were performed to address my previous questions and the overall quality of the article has improved. I recommend publication in its present form.

Depth-enhanced high throughput microscopy by compact PSF engineering

Author Response to Referees

We wish to thank the editor and reviewers for the attention paid to our revised manuscript. Reviewer comments are shown in black; our response is in blue. Extracts from the manuscript are in italics.

In the revised manuscript, added/changed text red colored.

Reviewer #1:

The authors have solved all my questions and confusions. I consider it could be published in nature communications.

We thank the reviewer for the positive opinion on our manuscript.

Reviewer #2 (Remarks to the Author)

The authors demonstrated adeptness in addressing the reviewer's concerns. The strategic relocation of supplementary explanations from the main paper to the supplementary information notably enhanced the article's clarity and accessibility. I commend the authors for taking this positive step.

We thank the reviewer for the appreciation of our revision.

Furthermore, I would appreciate it if the authors could consider addressing the following concerns to enhance the overall quality of the manuscript further.

- Regarding the question concerning the advantages of the mentioned methods over the light field, the authors briefly highlighted the lower lateral resolution of light field. However, with the recent advancements in deep learning-based post-processing computations for light field reconstruction, researchers have successfully achieved significantly improved lateral resolution compared to previous methods. Notably, even in light field deconvolution, the number of iterations can impact output resolution. Furthermore, in the context of 3D data, axial resolution holds significance. Therefore, it is crucial to inform readers about the advantages or disadvantages of the discussed methods compared to light field techniques.

We appreciate the reviewer's comment regarding comprehensive comparisons of our approach with other leading computational microscopy methods. Indeed, there is plenty of room for comparisons of such nature. We'd like to emphasize that the scope of this work is by introducing a novel way to implement PSF engineering in a minimally invasive way, in microscopy systems where it was not previously possible. In that sense, a thorough discussion of light field advancements would be misplaced, unless we conceive of a way to implement light field while still enabling an unmodulated usage of the microscope (such as replacing the turret). Our approach preserves the unmodulated performance of the microscope, which is an advantage – it enables peak sharpness for 2D without post-processing.

Naturally, there is always room for comparison between different microscopy approaches, however we feel that diving into resolution comparisons with other microscopy techniques that are not directly applicable in our setting would be beyond the scope of this manuscript.

We followed the suggestion by the reviewer and added a note regarding the fact that resolution of light field can indeed be improved by appropriate post-processing, with two relevant references:

“...,although resolution as well as reconstruction speed can be significantly improved using appropriate post-processing^{36,37} ”

36. Lu, Z. et al. Phase-space deconvolution for light field microscopy. *Opt Express* 27, 18131 (2019).

37. Wang, Z. et al. Real-time volumetric reconstruction of biological dynamics with light-field microscopy and deep learning. *Nat Methods* 18, 551–556 (2021).

- The authors mentioned deconvolution's impact on output quality but did not specify the algorithm used. Clarifying the deconvolution algorithm and discussing the effects of iteration variation on final resolution would enhance transparency and reader understanding of the methodology.

Deconvolution information is now added in the manuscript, stating that the algorithm used is Lucy Richardson. Information about the effects of iterations is now added in SI section 1. EDOF and standard PSF deconvolution:

“The number of iterations was set as 5. This value was selected as a trade-off between a number of iterations too low to achieve significant improvement of resolution, and an excessively high number of iterations that results in artifact generation.”

- The authors have not made simulation codes available online, which could improve the reliability and understanding of their data. I recommend they consider sharing the codes online for transparency and reproducibility.

All code, including the newly described EDOF deconvolution part, is now available online at <https://github.com/EliasNehme/HTPmicroscopy/>

Reviewer #3 (Remarks to the Author)

The authors have carefully revised the paper based on the comments provided by the reviewers. In particular, new experiments were performed to address my previous questions and the overall quality of the article has improved. I recommend publication in its present form.

We thank the reviewer for the positive opinion on our manuscript.

Reviewers' Comments:

Reviewer #2:

Remarks to the Author:

The authors addressed my comments.